# Electrical Discharges in Oil-Lubricated Rolling Contacts and Their Detection Using Electrostatic Sensing Technique

**DOI:** 10.3390/s22010392

**Published:** 2022-01-05

**Authors:** Kamran Esmaeili, Ling Wang, Terry J. Harvey, Neil M. White, Walter Holweger

**Affiliations:** 1nCATS, Faculty of Engineering and Physical Sciences, University of Southampton, Southampton SO17 1BJ, UK; kamran.esmaeili@soton.ac.uk (K.E.); t.j.harvey@soton.ac.uk (T.J.H.); walter.holwegerm@t-online.de (W.H.); 2Electronics and Computer Science, Faculty of Engineering and Physical Sciences, University of Southampton, Southampton SO17 1BJ, UK; nmw@ecs.soton.ac.uk

**Keywords:** electrostatic sensor, voltage measurement technique, electrical discharges, WEC formation, quantification algorithm, detection and diagnosis

## Abstract

The reliability of rolling element bearings has been substantially undermined by the presence of parasitic and stray currents. Electrical discharges can occur between the raceway and the rolling elements and it has been previously shown that these discharges at relatively high current density levels can result in fluting and corrugation damages. Recent publications have shown that for a bearing operating at specific mechanical conditions (load, temperature, speed, and slip), electrical discharges at low current densities (<1 mA/mm^2^) may substantially reduce bearing life due to the formation of white etching cracks (WECs) in bearing components, often in junction with lubricants. To date, limited studies have been conducted to understand the electrical discharges at relatively low current densities (<1 mA/mm^2^), partially due to the lack of robust techniques for in-situ quantification of discharges. This study, using voltage measurement and electrostatic sensors, investigates discharges in an oil-lubricated steel-steel rolling contact on a TE74 twin-roller machine under a wide range of electrical and mechanical conditions. The results show that the discharges events between the rollers are influenced by temperature, load, and speed due to changes in the lubricant film thickness and contact area, and the sensors are effective in detecting, characterizing and quantifying the discharges. Hence, these sensors can be effectively used to study the influence of discharges on WEC formation.

## 1. Introduction

Since the early 1900s, the detrimental effects of electrification on drive train components have been investigated [1,2,3,4], with respect to the downtimes caused by rolling element bearing failures under the presence of “shaft voltages”. The presence of shaft voltages has been shown to create a range of bearing damages such as electrical pitting, frosting, fluting, and corrugation [5]. Shaft voltages may be caused by winding faults, unbalanced supplies and electrostatic effects in electric motors and generators.

Research in the early years was focused on the influence of low-frequency shaft voltages on bearing life due to the popular use of DC and low-frequency AC machines in a variety of applications. The research led to a number of suitable mitigation techniques, such as high wear resistance brushes that maximize permissible residual shaft voltages [6,7,8,9,10,11].

With the introduction of variable speed drives in the 1980s, a substantial improvement was made in machinery efficiency. However, this has led to the presence of high-frequency shaft voltages on bearings and reappearance of fluting and corrugation damage. Subsequently, a number of new techniques were developed to effectively suppress both high-frequency and low-frequency AC as well as DC shaft voltages [12,13,14,15,16,17]. These included the use of an electrostatically shielded induction motor [14], equalization of ground potential with high-frequency bonding, reduction of inverter output, and lowering of the pulse width modulation carrier frequency [13].

Busse et al. [17] used experimental data in order to understand the influence of current density on the lifetime of the bearings and found that a theoretical model can be established to predict the lifetime of the bearing under the influence of electrical current. Accordingly, the expected theoretical bearing life may be adjusted by incorporating current density, at which it was found that current densities below 400 mA/mm^2^ are “safe”, as the bearing life exceeds 100,000 h (typical bearing life for industrial applications is typically 20,000–40,000 h) [17]. Beyond this “safe” zone, an increase of the current density can significantly reduce a bearing’s life, leading to early failures. It has been discussed in the literature that manifestation of electrical discharges, due to the presence of a shaft voltage and the dielectric properties of the lubricant, drives fluting and corrugation damages [18]. However, recent studies have revealed that at much lower current densities, e.g., <1 mA/mm^2^, bearings’ life has been significantly reduced (to 5–10% of the expected L_10_ life) due to the formation of WECs, but often combined with the presence of critical lubricants [19,20,21]. In these studies, it was concluded that the electrical discharges promote the formation of WECs [19,20,21]. As illustrated in Figure 1, the electrical circuit of a bearing race-lubricant-rolling element (a bearing contact) can be simplified as a capacitor in parallel with a resistor [19]. When the bearing contact is subjected to an electrical potential, charges will be accumulated on the race and rolling element surfaces, i.e., charging up. When the electric field reaches the lubricant’s breakdown strength, discharging occurs between the surfaces through the lubricant film.

Loos et al. [19] suggested that electrical discharging between bearing surfaces can lead to decomposition of lubricant molecules and subsequently electrochemical reactions at nascent surfaces forming hydrogen cations. Holweger et al. [22], however, suggested that electrical discharging may induce thermal effects at material inhomogeneities such as carbides or non-metallic inclusions, causing local thermal stress and strain that subsequently drives atoms such as carbon, chromium, and silicon to migrate and diffuse.

To date, limited monitoring techniques have been developed to detect electrical discharges, especially if they are transient. These techniques are often unable to detect local electrical discharges, suffer substantially from noise and require complex installation, especially in the industrial machinery and heavy equipment, where space is limited.

Electrostatic (ES) sensing techniques are proven to be capable of detecting charge-related phenomena, e.g., white layer formation [23], bearing failures in rolling contact fatigue mode [24], and WEC [20,21]. This study investigates electrostatic discharging on a twin-roller machine under various operating conditions where the applied electrical potential, lubricant temperature, entrain velocity, and load are varied, while the feasibility of ES sensing technique detecting discharging events is evaluated.

## 2. Materials and Methods

### 2.1. The Test Rig and Samples

Tests were conducted on a Phoenix Tribology Ltd. TE74 twin-roller machine, where two discs are loaded against each other using a pneumatic system, which can apply a radial load of up to 12 kN (see Figure 2). The lubricant is supplied through the inlet port to the discs, and the excess oil is collected from the oil sump to the tank. The temperature of the lubricant inside the tank can be adjusted for a temperature of up to 150 °C. The test rig utilizes three K-type thermocouples to monitor the temperature of the lubricant inside the tank and at the inlet and outlet ports. The electric motor of the bottom shaft can reach a speed of 3000 rpm, whilst the speed of the top motor can be adjusted using a 2-to-1 or 1-to-1 ratio to reach a speed of 6000 and 3000 rpm, respectively. The bottom disc was a crowned (transverse diameter of 100 mm) disc of 52 mm diameter, whilst the top disc was flat of 26 mm diameter.

### 2.2. Sensors and Data Acquisition Details

A DC power supply was used to apply a voltage between the top and bottom discs (see Figure 3). Brushes and sliprings were installed to provide a low-noise electrical connection to the shafts. A 50 kΩ resistor was used to limit the maximum current in the circuit, e.g., at a supply voltage of 15 V, the maximum current in the circuit is 300 µA.

Two ES sensors were utilized to investigate the local charges associated with charging and discharging events on the discs. The sensors’ faces were 6 mm in diameter and were positioned approximately 0.4 mm away from the disc surfaces (see in Figure 2b,c). To ensure the correct insulation and wiring of the electrostatic sensors, using a multimeter the electrical isolation of the sensor face and shield (as well as the outer body) were tested. The output from the electrostatic sensors was converted to voltage using charge amplifiers of type Brüel & Kjær 2635 (Darmstadt, Germany). The transducer sensitivity was set at 0.5 pC/V. The high-pass filter was set to 0.2 to avoid losing any low-frequency signals.

The voltage between the two discs and the outputs of the two ES sensors were recorded using PicoScope 4424, at a sampling rate of 5 MHz for 1 s at 20-s intervals throughout the tests. Additionally, the current in the circuit, measured by the voltage across the shunt resistor, and supply voltage were recorded using a separate data acquisition device at 10 kHz for 1 s at an interval of 20 s throughout the test duration. Rotational speed (of both discs), radial load, and temperatures (lubricant inlet, outlet, and tank) were recorded by the test rig software continuously at 1 Hz.

### 2.3. Testing Material

The test rollers were machined from SAE 52100 bearing steel bar that was austenitized at 850 °C and quenched in oil. Following this, the tempering process was performed at 180 °C for two hours. The chemical composition and mechanical properties of the standard SAE 52100 bearing alloy steel are shown in Table 1 and Table 2, respectively.

### 2.4. Lubricant and Additive Packages

Previous studies focused on exploring the critical lubricants that result in WEC formation and have identified a lubricant known as WW1362, as the most severe lubricant resulting in a WEC formation [26]. This lubricant called WW1362 is based on PAO with a viscosity of 46 mm^2^/s at 40 °C (ISO VG 46), including 5% viscosity index (VI) improver based on Polymethylmethacrylate (PMMA) and 2% Zinc dithiophosphate (a mixture of primary and secondary) and is used for the current studies on the TE74 twin-roller. The properties of the WW1362 lubricant are shown in Table 3.

### 2.5. Roughness Analysis

Surface roughness analysis was performed before the testing by using a Talysurf 120 L laser profilometers from Taylor Hobson and according to ISO 4288-1996. The stylus was diamond-tipped with a radius of 2 μm and a vertical resolution of 0.25 nm. As the discs were polished in the rolling direction, they had a periodic pattern. Thus, a spacing distance (RSm) was used with an appropriate cut-off value. According to the standard, the evaluation length must be at least 5 times the cut-off value to give at least 5 evaluation windows for accurate measurement of the roughness. For all the analysis performed here, the evaluation length was kept constant at 9 mm and in the transverse direction, resulting in more evaluating windows for better accuracy.

Also, four equally-spaced measurements were performed for each disc to get an accurate estimation of the roughness, throughout the circumference. The pre-test average root mean square roughness values of the 26 and 52 mm disc were measured to be approximately Rq 0.025 µm and 0.029 µm, respectively and after the tests 0.032 for both discs.

### 2.6. Calculation of Lubricant Film Thickness and Lambda

For all the performed tests, the measurement of the slave and master shaft speeds, the lubricant inlet temperature, and the radial load, together with the initial roughness of the discs allowed for the calculation of minimum film thickness and lambda for the duration of the tests. The minimum film thickness and lambda are calculated using the equations detailed in Appendix A. Figure 4 shows the responses from the test rig sensors including the master and speed shaft speeds, the measured load at the contact, and various temperature measurements, as well as the calculated contact area, minimum film thickness, and lambda for each measurement.

### 2.7. Test Matrix

During testing, five parameters, rotation speed; load; lubricant inlet temperature; slide-to-roll ratio (SRR); and supply voltage, were varied to study their influences on the discharging events.

The test matrix using five parameters is detailed in Table 4. For the investigation of the influence of supply voltage, 32 separate tests were performed, utilizing combinations of four speeds and four loads under two different electric current flow directions (bottom-to-top and top-to-bottom).

## 3. Results

### 3.1. Influence of Operating Parameters on VADC

Figure 5 shows the average VADC (this is calculated as single point for each of the 10,000 points captures, which occur every 20 s) from the first three types of tests, where the influence of disc rotating speed, load, and oil inlet temperature, at a constant supply voltage, are evaluated.

For the increase in the disc rotating speed, initially, no increase in average VADC is observed until the speed reached 450 rpm. Following this, a linear increase in average VADC is observed for the increase in rotating speed until reaching a speed of 2100 rpm. However, with any further increase in rotating speed, beyond the 2100 rpm until reaching a maximum speed of 3000 rpm, very little to no increase in average VADC is observed.

Increasing the load from 500 to 5570 N (contact pressure of 1.3–3 GPa) results in a decrease of 0.35 to 0.05 V for the average VADC (see Figure 5b). Initially, a rapid decrease is observed until reaching a load of 1500 N, followed by a gradual decrease with any further increase in load.

For the oil temperature test, initially, a rapid decrease of the average VADC is observed when the oil temperature increases from 40 to 60 °C (see Figure 5c). As the oil temperature continues to increase, the average VADC is reduced to almost zero until the temperature reaches 85 °C. Following this, a small increase in the voltage is observed.

### 3.2. Influence of Slide-to-Roll Ratio on VADC

As shown in Figure 6, under a constant load, rotating speed, oil inlet temperature, and supply voltage, when the SRR increases from 0 to 5%, the average VADC initially increases from 0.3 to 0.45 V. However, for any further increase of SRR between 5–15%, the average VADC shows a fluctuating response with a variation in the median value of 0.4–0.45 V. It is worth mentioning that for each SRR step (lasting 15 min), the variation between the first and the third quartiles (bottom and top horizontal line of the box), is less than 0.04 V.

### 3.3. Influence of Supply Voltage on VADC

Figure 7a presents the results from two extreme cases, where the highest and lowest average VADC are seen, whilst Figure 7b presents 8 of the 32 cases.

As observed in Figure 7a, for the 3000 rpm 1145 N case, as the supply voltage increases, a logarithmic growth or second-order polynomial response is observed for the average VADC, i.e., a rapid initial increase followed by a reduction in the rate of increase until a plateau occurs. This logarithmic behavior is also observed for all the 32 conducted tests, although for some of the tests with lower average VADC, the logarithmic response is less pronounced. This can be seen in Figure 7b, showing 8 representative tests from the 32 tests, comprised of all the speeds and the highest and lowest loads (8 tests).

Besides, it has become clear that the peak of average VADC is influenced by the rotational speed and load, as tests with higher speeds and lower loads reach higher average VADC for the increase in supply voltage.

### 3.4. Quantification of Electrical Discharges

Figure 8 shows the average VADC as a function of the increase in disc rotating speed for the test presented in Figure 5a and two repetition tests, and four snapshots of the VADC (transformed using a low-pass filter with a cut-off frequency of 300,000 Hz to see the electrical discharges). Each of the snapshots shows the voltage responses of 100 µs out of the time frame of one second, to see the individual electrical discharges. At 0 rpm, the average VADC is at approximately 0 V and no electrical discharges can be seen in Figure 8a. Despite the average VADC showing approximately 0 V at 0 rpm, the contact can show a very small resistance due to the material resistance of the discs, shafts, sliprings/brushes pairs, and oxide layers at the contact as well as the resistance of the electrical circuit. These together should impose a total resistance of a few ohms, which are not significant compared to the 50 kΩ regulating resistor in series. Thus, showing approximately 0 V. At 1000 rpm, one “peak” can be seen in Figure 8b, representing one charging–discharging event, whilst at 2000 rpm (see Figure 8c) three discharges at higher voltages are seen. Similarly, at 3000 rpm, the number of discharges increases significantly with some overlapping events. For all the tests conducted as a part of the fundamental studies, a similar relationship was seen between the average VADC and the number and amplitude of electrical discharges. Looking at the snapshot at 2000 rpm in more detail in Figure 9, each of the three peaks includes a charging period (slowly rising) and a rapid discharging period (sharp decreasing). The amplitude of a discharge event generally varies depending on the charging time, i.e., the longer the charging time, the higher the peak value.

As illustrated in Figure 10, to understand the characteristics of the charging–discharging phenomenon at the contact, under different operating conditions and the influence of supply voltage, a statistical method has been developed to quantify the number and intensity of the charging–discharging events that occurred during each test.

All charging–discharging events above a threshold are automatically identified (based on the rise and fall of signal characteristics) and the voltage is recorded as the discharge amplitude. The threshold is set to 0.07 V, considering the level of baseline and noise, whilst allowing for discharges as small as 0.07 V to be identified. The measured discharge amplitudes are then recorded into a table with their given identification number for the duration of the tests. Following this, at each instant of time, the electrical discharges are grouped according to their discharge amplitude into multiple discharge bin groups, and their occurrences in each bin are counted inside a secondary table. Hence, enabling a semi-quantitative analysis of the number and amplitude of discharge events throughout the tests using a visual representation. Knowing the operating parameters’ information for each instant of time (i.e., at 20 s), the time column can be replaced by the corresponding operating parameter (as indicated in Figure 10, where time is replaced by rotational speed).

As shown in Figure 11a, for the disc rotating speed test, at the beginning of the test, a small number of low-amplitude discharging events are observed. After a speed of 1000 rpm is reached, a small number of high-amplitude discharges are seen. As the disc speed increases, the intensity of medium amplitude events increases until the end of the test.

Overall, whilst only changing one operating parameter at a time, an increase in the disc speed leads to high-amplitude events as well as an increasing number of electrical discharges. An increasing load applied to the contact leads to a lower number of discharging events and lower amplitude discharges. Increasing temperature initially reduces the number and amplitude of discharges to a minimum level; however, above 85 °C, the number and amplitude increases. Increasing the supply voltage increases the number and amplitude of electrical discharges. It is worth noting that the maximum discharging amplitude observed in these short tests is 1.50 V.

### 3.5. Detection of Electrical Discharges Using Electrostatic Sensing Technique

Figure 12 shows two sets of sensors’ responses, a and b from two identical tests, but the electric current supplied to the contact was reversed in one. Figure 12a shows the responses from the sensors when current flowed from bottom-to-top discs, whilst Figure 12b illustrates the responses with a current from top-to-bottom discs.

As shown in Figure 12a,b, the electrical discharges detected in the VADC signals are also present in the signals from the ES sensors, and they appear synchronized with the VADC in time, same shape, and amplitude, but in the opposite polarity. However, the discharges in each case are only detected using one of the electrostatic sensors, whilst the other shows approximately a flat response and they appear to be dependent on the direction of the current.

Figure 13 shows the number and amplitude of electrical discharges detected in the absolute signals from the bottom ES sensor as a function of those in the VADC, for the disc rotation speed, load, and oil inlet temperature tests. As shown in Figure 13a, throughout the tests, the number of discharges detected in the ES sensor’s signals monitoring the bottom disc increases approximately linearly with the increase in the number of discharges detected in the VADC. This is further confirmed by the linear smoothness line that is implemented for the detected number of discharges for the tests.

Figure 13b shows the average amplitude of discharges detected in the absolute signals of the ES sensor monitoring the bottom disc as a function of those detected in the VADC. As the data points and the linear smoothness line indicate, the average amplitude of discharges detected by the ES sensor monitoring the bottom disc increases approximately linearly with the increase in the average amplitude of discharges detected in the VADC. For the shown cases, although approximately a linear relationship can be established, small variations exist between the ES sensor and VADC signals. While limited data is presented, all the other data exhibit a similar relationship.

## 4. Discussion

### 4.1. Influence of Operating and Electrical Parameters on the VADC

It has been shown that the disc rotating speed, load, temperature, SRR, and supply voltage all influence the average VADC. For rotational speed, initially 0 V was recorded. This is due to zero or very low entrainment velocities resulting in boundary lubrication producing metal-to-metal contact, leading to an effective short-circuit. As the speed increases, the entrainment velocity increases and the discs begin to separate and insulating dielectric allows a potential to develop [27,28]. However, the rate of increase in average VADC was reduced with further increase in speed, which can be associated with a non-linear relationship (see Figure 14a) between the rotating speed and the minimum film thickness (see Appendix A, where the rate of increase in both minimum film thickness and lambda reduces with an increase in speed). As the electrical discharges are always active in the lubricant, a less resistive pathway is often created through the lubricant, which can facilitate the discharge of the charges through the lubricant. Thus, this might have also partially prevented the average VADC to increase proportionally with the increase in the oil minimum film thickness [29] (pp. 74–77).

The increase in load, despite a relatively minor influence on the minimum film thickness, substantially reduced the average VADC. Recalling Figure 7 from [19], the increase in radial load, also substantially increases the contact area (see Figure 14b showing a change in the contact area from 0.7 mm^2^ to 2.8 mm^2^). Thus, the decrease in average VADC may be a combination of a greater contact area that increases approximately linearly with the increase of the load and a decrease of the oil film thickness [27,28], which lead to a reduction of electrical resistance at the discs’ contact [27,28].

An increase in the lubricant temperature also reduced the average VADC to approximately 0 V until reaching a temperature of 85 °C, from which a small increase in the average VADC is observed. The initial decrease in the average VADC can be similarly explained by the substantial decrease in the oil film thickness (see Figure 14c showing a decrease in lambda from 14 to 3.5) [27,28] and/or a change in the movement of the charged species within the lubricant, which can facilitate the movement of the charge across the contact. However, the increase in the average VADC beyond 85 °C is not driven by a change in the physical (contact area or lubricant film thickness) properties. This may have been influenced by a partial formation of a tribofilm on the mating surfaces, which can have a greater electrical resistance [30,31,32] or a change in the movement of the charge through the lubricant (i.e., charge relaxation [33]).

The increase in SRR from 0 to 15% initially resulted in a rapid increase in the average VADC from 0.3 to 0.45 V, followed by a fluctuating response around 0.45 V for any increase in SRR beyond 5%. Although a variation in SRR changes the experienced lambda (see Figure 14d showing a decrease in lambda from 15 to 14), in this study it was found that the change was too small to have any major influence on the average VADC. Also, the reduction in lambda is shown to decrease the average VADC in the disc rotating, load and temperature tests. Thus, it is not clear what mechanisms have resulted in a higher average VADC.

Across all the 32 tests for the supply voltage, it was observed that the tests performed under higher rotating speeds and lower loads reach a higher average VADC, which further confirms the results obtained, where the increase in rotating speed increased the average VADC, whilst the increase in load, decreased the average VADC. Also, across all the tests, with a linear increase of supply voltage (from 0 V to 15 V), a logarithmic growth or second-order polynomial response for the average VADC was observed. The rate of increase in the average VADC nearly flattened for any further increase of supply voltage above 10 V (200 μA), which might demonstrate the same voltage-current characteristics as those reported by Loos et al. [19] and Zuercher et al. [34]. This voltage-current characteristic is driven by the dependency of the contact’s electrical resistance on the electric current through the contact(s). Any increase in the level of current (and faster transfer of charge across the contact), leads to the presence of a low-resistance pathway through the lubricant and subsequently lower electrical resistance and VADC.

### 4.2. Characteristics of Electrical Discharges

Using high-resolution windows, it was shown that electrical discharges with different amplitudes can exist in singular form and/or as an overlapping network. An electrical discharge occurs when streamers find the narrowest and greatest conductive pathway to discharge within the lubricant by overcoming the dielectric strength of the lubricant [35]. In a tribological contact, this is through the asperities where the distance between the asperities of the mating surfaces is small. This corresponds to the highest local electric field where the charges can overcome the dielectric strength of the lubricant. In the case of WW1362 lubricant with a dielectric strength of 26.4 kV/mm and a voltage drop of 1 V across the contact, a local film thickness of 0.037 µm is sufficient to overcome the dielectric strength of the lubricant.

Thus, it is vital to consider all the parameters that influence the presence of an electrical discharge. These parameters are the voltage across the contact, the local lubricant film thickness, the local contact area, and the dielectric strength of the lubricant. Figure 15 shows how these electrical discharges are influenced by the instantaneous condition of the contact and asperities (assuming no change in the quality of the lubricant). As Case 1 shows, high asperity to asperity contacts lead to the discharge of the voltage through the metal-to-metal contacts. As the resistance between the metal-to-metal contact is low, it appears as approximately zero V. However, as shown in Case 2, charges start to accumulate across the contact when the surfaces are momentarily separated from each other, leading to the charging period. Case 3 shows the instance where the accumulated charges discharge in less than 2 μs when asperities come close to each other, reducing the electrical resistance. It is worth mentioning that in Case 3, the charges do not fully discharge across the contact (see the voltage trend), which is due to the asperities coming close to each other, but not close enough to facilitate a metal-to-metal contact. On the other hand, Case 4 shows a moment when the asperities form a metal-to-metal contact with each other. This results in a full discharge of the accumulated charges across the contact and a reduction of voltage to approximately 0 V.

It is worth mentioning that in the case of identical local lubricant film thicknesses, a higher local contact area between the two mating surfaces leads to a lower local resistance (see Figure 7 in [19]) due to the relatively larger surfaces in contact where streamers can discharge.

The other parameter is the dielectric strength of the lubricant, which can be represented as a combination of capacitor and resistor (see Figure 1). For a purely analytical model, the combination of capacitor and resistor can be embedded into the existing electrical circuit of the TE74 test rig (see Figure 3) to analyze the influence of lubricant dielectric on the characteristics of electrical discharges. Considering a supply voltage of 10 V and a regulating resistor of 50 kΩ, a variety of lubricant resistance and capacitance combinations can be analyzed (see Figure 16) [36] (The resistor-capacitor (RC) circuit in series with regulating resistor can be solved according to [36] (p. 232) to show the charge on the capacitor as a function of time). Under a constant capacitance of 100 pF, the increase in the resistance of the lubricant increases the maximum amplitude of the electrical discharge (see Figure 16a). As shown in Figure 16b, under a constant resistance of 100 kΩ, the increase in the lubricant capacitance reduces the rate of increase in the VADC. So, the lubricant resistance controls the amplitude of discharges whilst capacitance governs the slop of a discharge event.

However, the electrical parameters of the lubricant can be influenced by other parameters including lubricant consistency, impurities and their nature, types of additives, as well as the applied voltage, and the time over which the voltage is applied to the lubricants [29] (pp. 1–9). Surface contaminant and debris due to oxidation, deterioration, and aging of the oil can also influence the electrical behavior of the contact [37,38].

### 4.3. Electrical Discharges and Their Detection Using Electrostatic Sensing Technique

Using high-resolution windows, it was shown that the VADC is driven by the presence of electrical discharges. As the resultant average VADC increases, the discharges appear at higher amplitudes and occur more often (see Figure 8). Also, compared to the average VADC, the amplitudes of individual discharging events are much higher (1.5 V, as opposed to the maximum average VADC of 0.3 V, observed in Figure 11).

For the cases shown in Figure 11, when the average VADC reaches 0.3 V (high average VADC in these tests), the discharge bins of 0.3–0.5 V appear as the dominant electrical discharge bins, representing the highest number of electrical discharges of up to 16,000, whilst discharge amplitudes as high as 1.4–1.5 V are also introduced. However, as soon as the operating conditions reduce the average VADC to 0.1 V, the dominant discharge bins shift towards the discharge bin of 0.1–0.2 V, demonstrating the transformation of high-amplitude electrical discharges towards low-amplitude electrical discharges.

Electrical discharges, detected using the voltage measurement technique, also appeared in the signals from one of the ES sensors, depending on the direction of the current (see Figure 12). The reason that only one of the ES sensors detected the discharging events is due to the grounding of the electrical circuit. As one of the discs was always grounded in the current setup, this prohibits the accumulation of any charge on the disc, thus no discharges appear in the corresponding ES sensor.

The quantification algorithm was used for both voltage measurement and ES sensors’ signals and it was demonstrated that the ES sensing technique is effective in monitoring the same electrical discharges using a non-contact approach. It is worth mentioning that differences exist between the two monitoring methods, mainly due to the sensitivity of the two monitoring techniques, the sensors’ baseline noise, the threshold specified in the quantification algorithm (see Figure 10), and the gap between the ES sensor face and the disc.

## 5. Conclusions

This study has focused on investigating electrical discharge events in an oil-lubricated rolling contact under low current density conditions. Rolling contact fatigue tests under a range of loads, speeds, and oil temperatures were performed on the TE74 twin-roller test rig to study the influence of operating parameters on the electrical discharges between the two discs. In addition to voltage and current monitoring, ES sensors have been used during the tests and their feasibility of monitoring discharging events has been investigated. The main conclusions from this study are:Changing the contact conditions, e.g., load, speed, and oil temperature alters the average VADC in different ways. Increase in speed results in an increase in the VADC, while an increase in the load and oil temperature both lead to a reduction in the VADC. These changes are proposed to be driven by the change in the lubricant film thickness and contact area.Under all load and speed conditions, an increase of supply voltage initially leads to a rapid increase in the average VADC, until eventually reaching a plateau.Through high sampling rate data, it was found that operating parameters and supply voltages as well as influencing the average VADC, also influence the number and the amplitude of electrical discharges.Electrostatic sensors are found to detect the discharging events occurring on one of the discs (non-grounded) similar to that of the VADC. As electrostatic sensors have simple design and do not require the electronics to be embedded onto the sensors, they can be used in an oil-lubricated environment and under harsh conditions to monitor electrical discharges and diagnose WEC events.

## Figures and Tables

**Figure 1 sensors-22-00392-f001:**
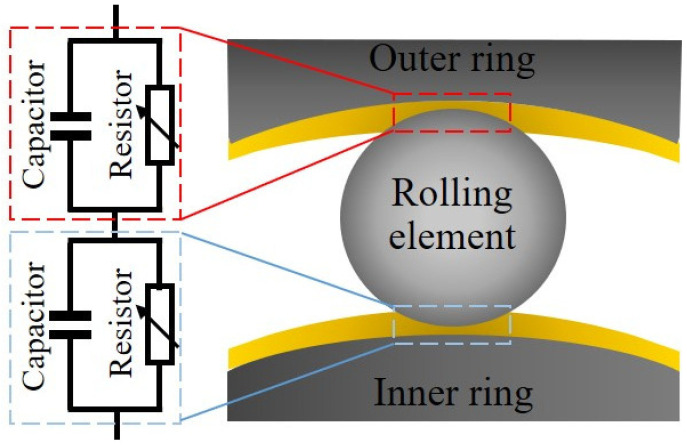
An illustration of simplified electrical circuits of race-rolling element-race contacts in a rolling element bearing lubricated by a dielectric lubricant.

**Figure 2 sensors-22-00392-f002:**
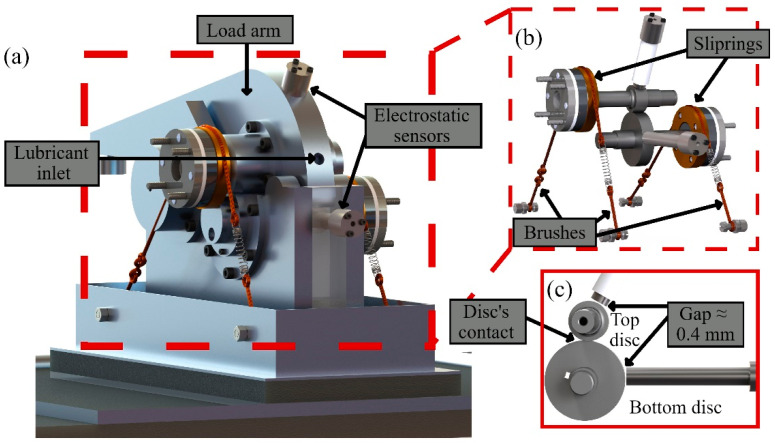
(**a**) A 3-D model of the TE74 twin-roller machine, showing the applied radial load using a load arm and brushes/sliprings installed to provide suitable electrical contacts for the rotating parts. (**b**) A 3-D model showing the top and bottom discs and positions of the two ES sensors. (**c**) A cross-section view of the model showing the gap between the ES sensors and the discs at approximately 0.4 mm and the discs’ contact.

**Figure 3 sensors-22-00392-f003:**
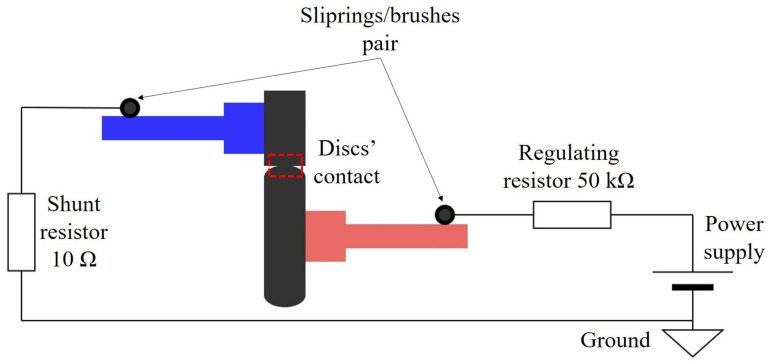
A schematic of the electrical circuit in the TE74 test rig.

**Figure 4 sensors-22-00392-f004:**
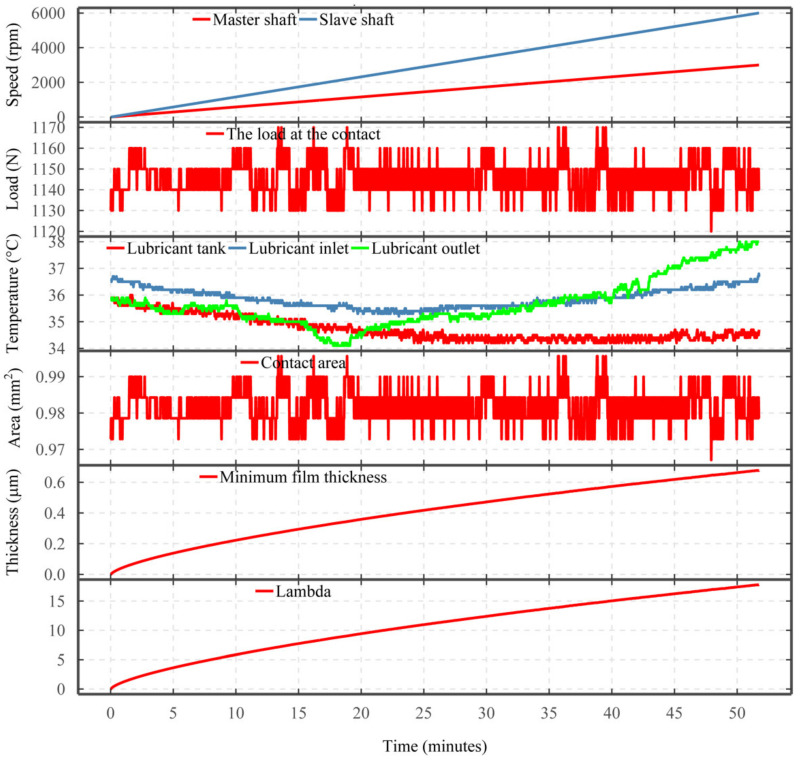
The responses from the test rig sensors and the calculated contact area, minimum film thickness, and lambda based on the initial roughness measurement.

**Figure 5 sensors-22-00392-f005:**
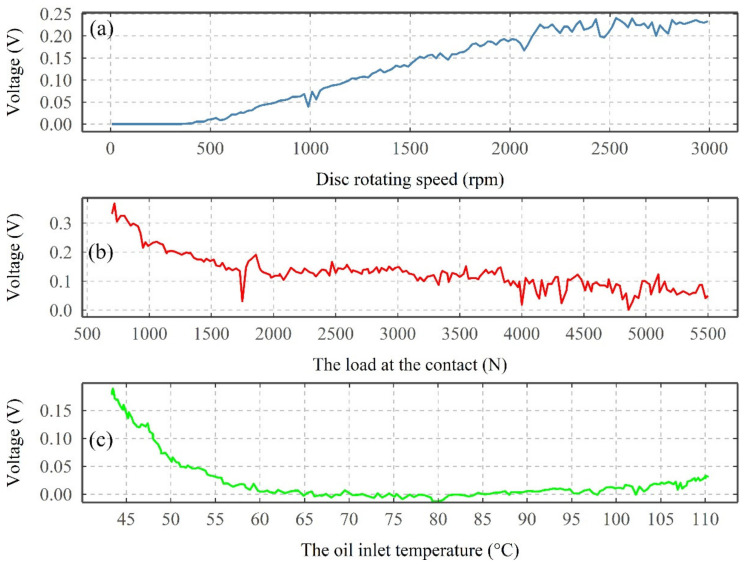
The average VADC under the influence of changing (**a**) the disc rotating speed, (**b**) the load at the contact, and (**c**) the oil inlet temperature.

**Figure 6 sensors-22-00392-f006:**
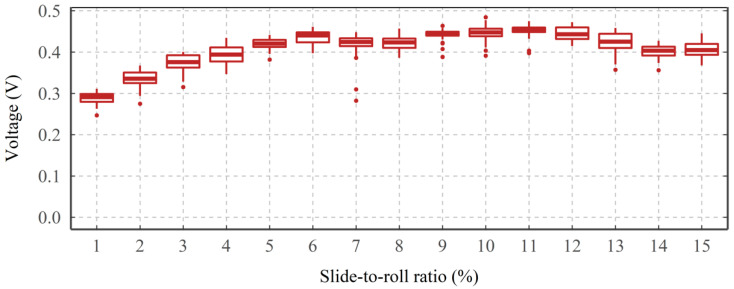
The average VADC for a variety of slide-to-roll ratios.

**Figure 7 sensors-22-00392-f007:**
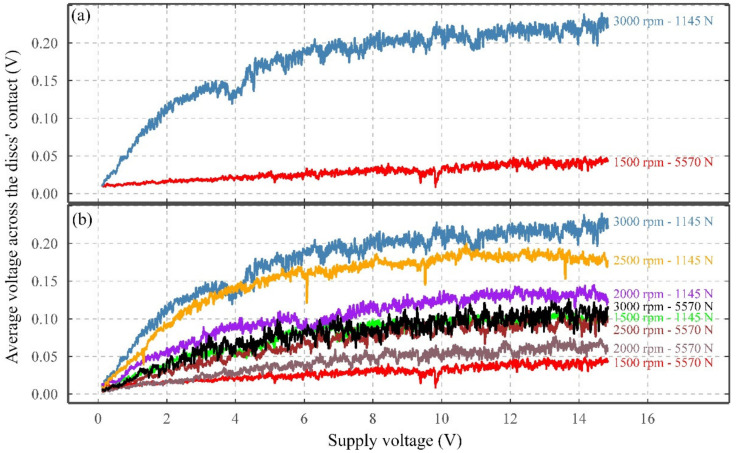
Influence of supply voltage on the average VADC for (**a**) two cases with the highest and lowest resultant average VADC and (**b**) selected cases with the highest and lowest loads, and all the speeds.

**Figure 8 sensors-22-00392-f008:**
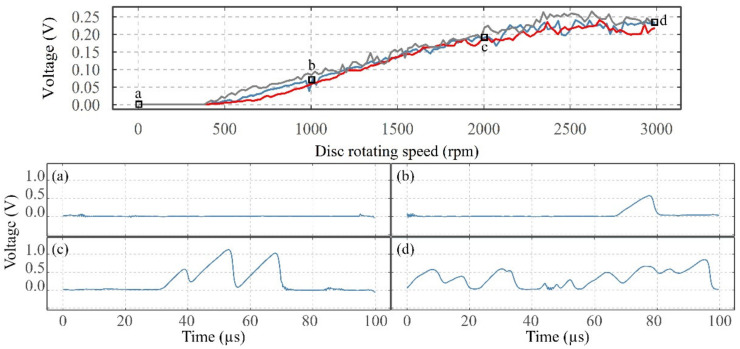
The average VADC as a function of a variation in speed and the snapshots of the voltage sampled at 5 MHz; (**a**) at 0 rpm, (**b**) at 1000 rpm, (**c**) at 2000 rpm and (**d**) at 3000 rpm.

**Figure 9 sensors-22-00392-f009:**
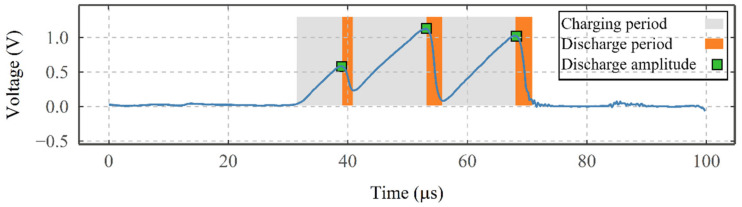
The charging and discharging periods, and the maximum discharge amplitude observed within the signals from VADC, in the speed test and captured at 2000 rpm.

**Figure 10 sensors-22-00392-f010:**
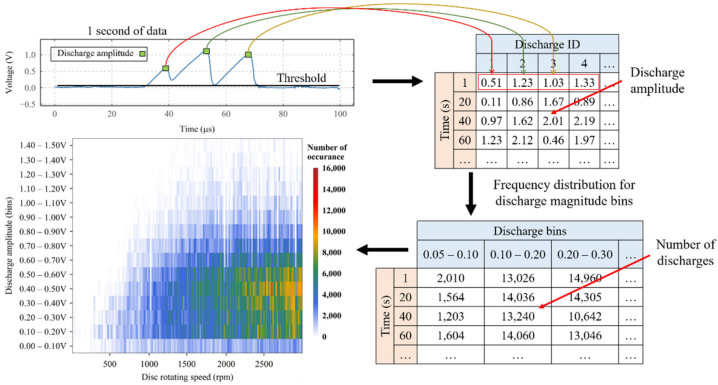
Schematics of the quantification algorithm developed in this study to evaluate the frequency distribution of electrical discharges in multiple discharge bin amplitude.

**Figure 11 sensors-22-00392-f011:**
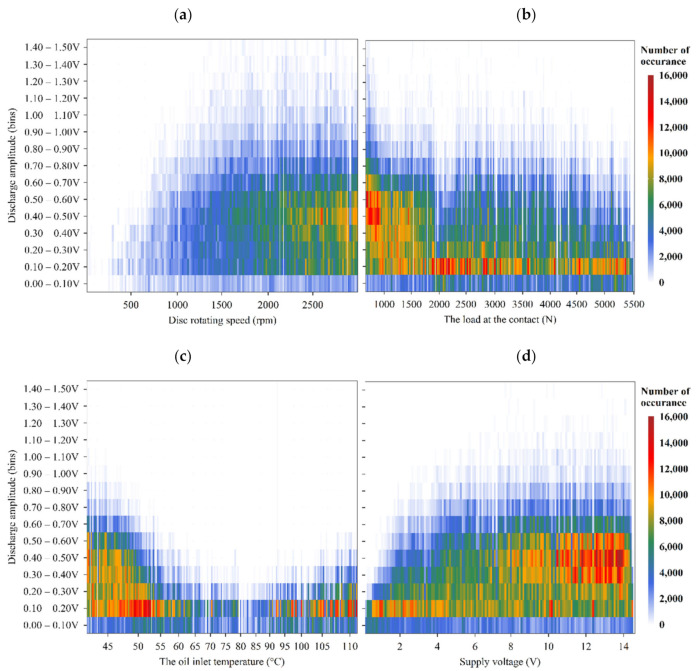
Color maps of the number of discharge events and their corresponding discharge amplitude (within defined bins) as a function of a change in test conditions; (**a**) disc rotating speed, (**b**) load applied to the discs’ contact, (**c**) lubricant inlet temperature, and (**d**) supply voltage under 3000 rpm, 1145 N and 40 °C conditions.

**Figure 12 sensors-22-00392-f012:**
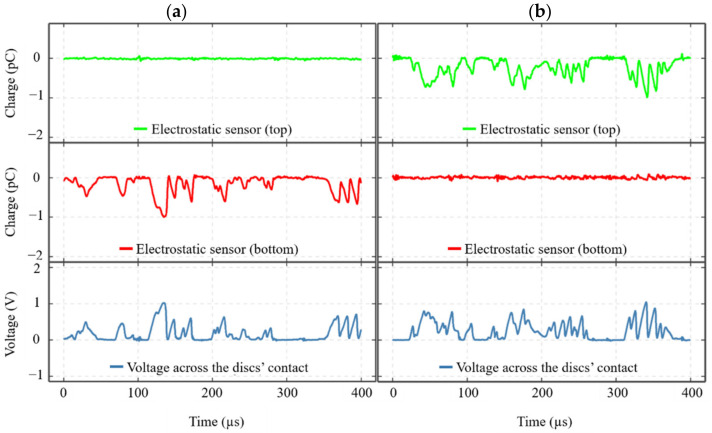
Discharging events detected by ES sensors and voltage measurement across the contact, from two tests conducted at a rotational speed of 3000 rpm, a load of 1145 N and lubricant inlet temperature of 40 °C whilst the supply voltage was varied. The data shown in the graphs were captured when the supply voltage was at 10 V. (**a**) The current flowed from the bottom-to-top disc, and (**b**) from the top-to-bottom disc.

**Figure 13 sensors-22-00392-f013:**
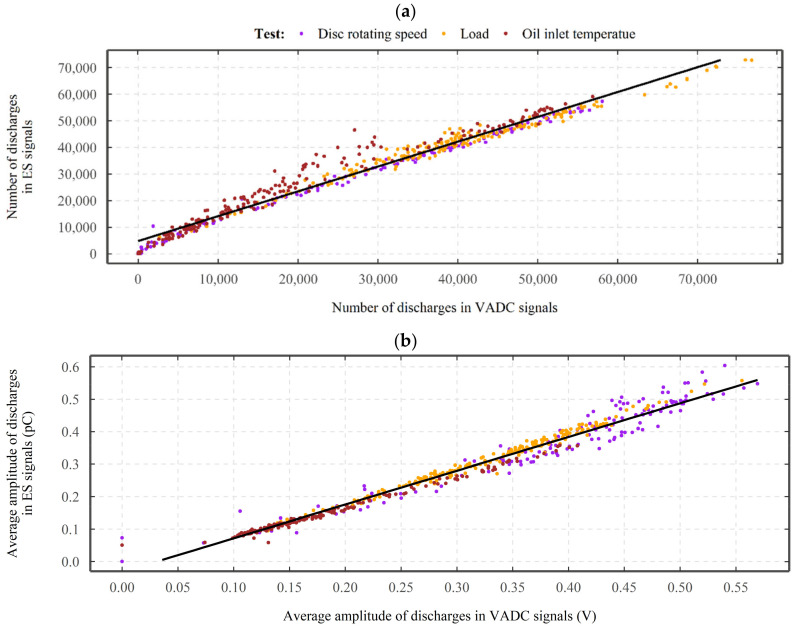
(**a**) Number and (**b**) amplitude of discharges, detected in the bottom ES sensor’s signals as a function of those detected on the VADC signals, for the disc rotating speed (magneta), load (yellow), and oil inlet temperature (red) tests.

**Figure 14 sensors-22-00392-f014:**
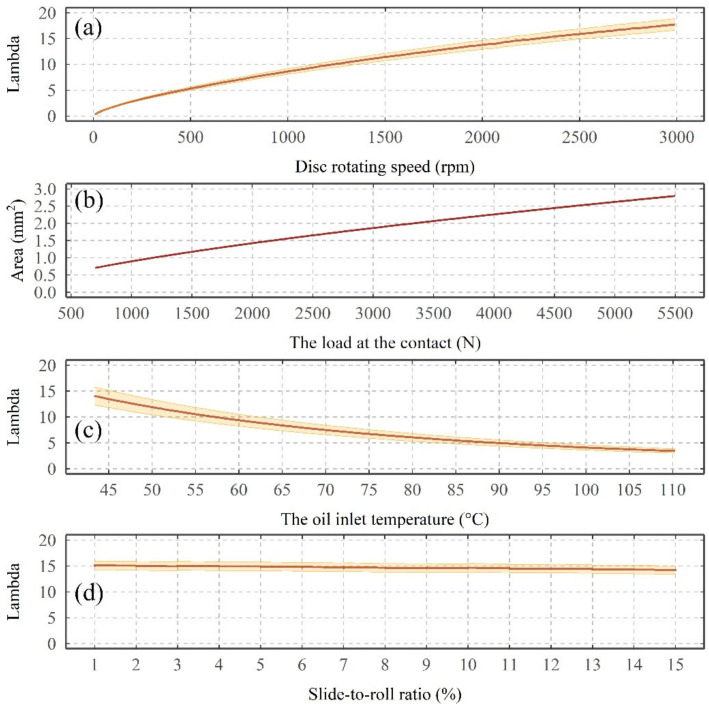
Influence of (**a**) disc rotating speed, (**c**) the oil inlet temperature and (**d**) slide-to-roll ratio on the experienced lambda. The ribbon area marks the calculated lambda based on the initial and the final roughness measurements. (**b**) Changes in the experienced contact area as a function of the load at the contact.

**Figure 15 sensors-22-00392-f015:**
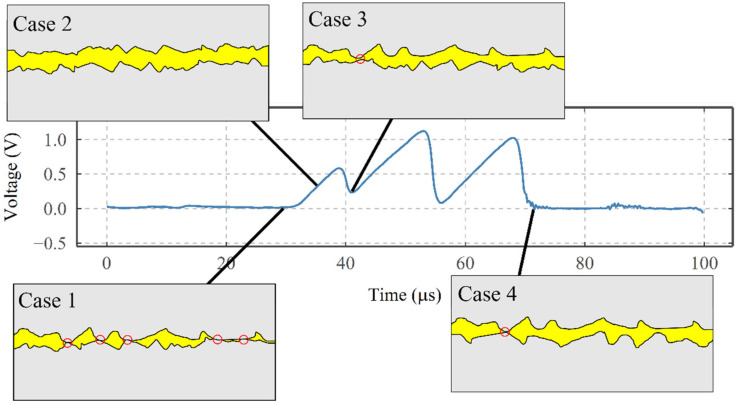
The charging and discharging periods together with the schematics, showing the influence of contact on the presence of these discharge events.

**Figure 16 sensors-22-00392-f016:**
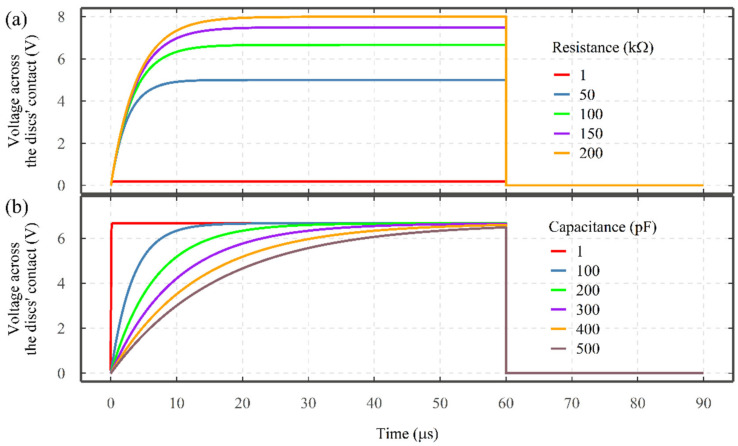
The influence of lubricant (**a**) resistance and (**b**) capacitance on the electrical discharge. Operating conditions = Supply voltage of 10 V, regulating resistor of 50 kΩ, charging period of 60 µs and (**a**) lubricant capacitance of 100 pF and (**b**) lubricant resistance of 100 kΩ.

**Table 1 sensors-22-00392-t001:** Chemical composition of SAE 52100 alloy steel [25].

Element	Content (%)
Iron, Fe	96.5–97.32
Chromium, Cr	1.35–1.60
Carbon, C	0.93–1.05
Manganese, Mn	0.25–0.45
Silicon, Si	0.15–0.35
Sulfur, S	≤0.015
Phosphorous, P	≤0.025

**Table 2 sensors-22-00392-t002:** Mechanical property of SAE 52100 alloy steel [25].

Properties	Metric
Bulk modulus	140 GPa
Shear modulus	80 GPa
Elastic modulus	210 GPa
Poisson’s ratio	0.27–0.30
Density	7.81 g/cm^3^
Melting point	1424 °C

**Table 3 sensors-22-00392-t003:** The physical properties of the WW1362 lubricant.

Properties	Value	Unit
Density at 15 °C	883	kg/m^3^
Dielectric strength	26.4	kV/mm
Kinematic viscosity at 40 °C	60.34	mm^2^/s
Kinematic viscosity at 100 °C	8.69	mm^2^/s
Pressure–viscosity coefficient	8	GPa^−1^

**Table 4 sensors-22-00392-t004:** Details of the tests; ^1^ incremental rate of 1 rpm/s, ^2^ incremental rate of 0.54 N/s, ^3^ approximate incremental rate of 0.01 °C/s, ^4^ 15 min for each 1%, ^5^ incremental rate of 0.1 V/s.

Influence of	Bottom Disc Speed (rpm)	Load/Contact Pressure (Max) (N/GPa)	Oil Inlet Temperature (°C)	Slide-to-Roll Ratio (%)	Supply Voltage (V)	Maximum Current (µA)
Disc rotating speed	0–3000 ^1^	1145/1.75	40	0	10	200
Load	3000	500–5570/1.3–3 ^2^	40	0	10	200
Oil temperature	3000	1145/1.75	40–110 ^3^	0	10	200
Slide-to-roll ratio	3000	1145/1.75	40	0–15 ^4^	10	200
supply voltage	1500	1145/1.75 1720/2.00 3340/2.50 5570/3.00	40	0	0–15 ^5^	0–300
	2000	as above		0		
	2500	as above		0		
	3000	as above		0		

## Data Availability

All data will be made available on request to the correspondent author’s email with appropriate justification.

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
