# Peer review of "Electrical Discharges in Oil-Lubricated Rolling Contacts and Their Detection Using Electrostatic Sensing Technique"

_sensors, 2022, doi:10.3390/s22010392_

Round 1

Reviewer 1 Report

Using voltage measurement and electrostatic sensor, this paper studies the discharge phenomenon and rolling contact fatigue phenomenon between lubricated steel rolling contact points of TE74 twin-roller machine under electrical and mechanical conditions, which is helpful to quantify the reliability of rolling bearing operation on site, but it needs some improvement before acceptance and publication. My detailed comments are as follows:

  1. The abstract does not explain how the speed, load and temperature specifically affect the level and times of discharge. It should focus on the innovation and contribution of the paper, and explain the different characteristics of the signals obtained by the electrostatic sensor under different working parameters, so as to prove the feasibility of the electrostatic sensor in the actual working conditions.
  2. Figure 4 in Section 2.6 shows that the test has only been carried out for 50min or only 50min data have been collected. Is this too short for rolling bearings under long-term operating conditions? Should the test time be extended to verify the credibility of this test method? For the verification of rolling contact fatigue, is 50 min wear test enough to make the rolling bearing reach its fatigue state?
  3. As for the influence of working parameters on average VADC in Section 3.1, when considering the influence of a single factor on VADC, whether the influence of two other factors on VADC should be considered.
  4. In Section 3.4 quantification of discharge, figure 8 shows the voltage change curve with the increase of rotating speed. Does the voltage change curve show the same effect in each test under the same rotating speed? The analysis of only one test result may not be rigorous, and more detailed description may be required for this section.
  5. As for the charging cycle and schematic diagram in Figure 15 in Section 4.2, whether the principle can be further explained more intuitively by observing the micro morphology by scanning electron microscope, and the state and related mechanism between the two contact surfaces in this part may need to be further described.
  6. The conclusion should be concise and only summarize the innovative and most important contributions in the research.

Author Response

Dear Reviewer 1,

Many thanks for your prompt and comprehensive feedback. Please see the attachment.

I hope you find the revised version of the manuscript satisfactory.

Kind regards,

Kamran

Reviewer 2 Report

This paper discusses the electrical discharging of rolling bearing lubricated by a dielectric lubricant. One test rig was set up to do the testing and results obtained are analyzed and discussed. Overall, the paper is written well and it is well organized.

Review comments are given below:  

Page 8, Line 209: It is Figure 6 not Figure 6A.

Through the text, it is suggested using the format like Figure 5a, Figure 5b, Figure 5c not Figure 5A, Figure 5B and Figure 5C, for example.

On Page 17, A) and B) on Figure 16 should be marked as a) and b).

Page 17, the bottom line: Please check if it is Figure 1.

On Figure 16 (Page 17): Each curve is increasing with time. Can each curve approach a constant value when the measurement time is longer enough? Please plot again to show if there exists a maximum and constant value or if each curve becomes a unimodal one finally.    

It is suggested to add a brief discussion on the usefulness of the electrical discharge detection technique developed in this paper. For example, it could be applied to bearing load measurement, operation and condition monitoring, etc. and if the technique can be easily implemented for field application.  

Author Response

Dear Reviewer 2,

Many thanks for your prompt and comprehensive feedback. Please see the attachment.

I hope you find the revised version of the manuscript satisfactory.

Kind regards,

Kamran

Round 2

Reviewer 1 Report

The author used multivariate experimental design methodologies to evaluate the effect of the variables, obtaining statistically significant results. The paper has certain novelty and advantages for this field research work and has value for publishing in Sensors.

Considering that this article uses electrostatic sensors to perform fault diagnosis tests with good results, and the article focuses on basic research. I suggest this manuscript can be published.

It is recommended that the mechanism behind the experimental phenomenon should be explained in-depth and the multi-factor coupling effect research should be carried out in the next work.

This manuscript is a resubmission of an earlier submission. The following is a list of the peer review reports and author responses from that submission.

Round 1

Reviewer 1 Report

The manuscript is well organized. The experimental setup is adequately explained. The results of numerous experiments as well as the sensor used are explained in detail. I think the manuscript is acceptable in its current form.  Although the manuscript is acceptable in its current form, it could be improved by adding a paper structure at the end of the introduction, taking into account the length of the paper itself.